# Lean Management in Medium-Sized Oral Cavity Defect Reconstruction: Facial Artery Musculomucosal Flaps Versus Free Flaps

**DOI:** 10.3390/jcm14248760

**Published:** 2025-12-11

**Authors:** Michał Gontarz, Emilia Lis, Konrad Biel, Jakub Bargiel, Krzysztof Gąsiorowski, Kamil Nelke, Dayel Gerardo Rosales Díaz Mirón, Grażyna Wyszyńska-Pawelec

**Affiliations:** 1Department of Cranio-Maxillofacial Surgery, Jagiellonian University Medical College, Poland University Hospital, 30-688 Cracow, Poland; jakub.bargiel@uj.edu.pl (J.B.); krzysztof.gasiorowski@uj.edu.pl (K.G.); grazyna.wyszynska-pawelec@uj.edu.pl (G.W.-P.); 2Students’ Scientific Group, Department of Cranio-Maxillofacial Surgery, Jagiellonian University Medical College, 30-688 Cracow, Poland; emilia.lis@student.uj.edu.pl (E.L.);; 3Maxillo-Facial Surgery Ward, EMC Hospital, Pilczycka 144 Street, 54-144 Wrocław, Poland; 4Instituto de Seguridad y Servicios Sociales de los Trabajadores del Estado (ISSSTE), Hospital Regional de 14 Torreon, Hospital Regional de Torreón, Torreón 27000, Mexico; cmfdayel@gmail.com

**Keywords:** oral cancer, oral cavity reconstruction, soft tissue free flaps, facial artery musculomucosal flaps, FAMM flaps, radial forearm flap, RFFF

## Abstract

**Background/Objectives**: Oral cavity defects highlight the necessity for individualized reconstructive strategies given their anatomical and functional complexity. Reconstructive surgery should optimize healing and function, minimize complications, and reduce operative time and costs. Although free flaps remain the gold standard for oral cavity reconstruction, they require large teams, specialized microsurgical equipment, and extended operative times. **Methods**: A retrospective analysis was performed on 59 consecutive patients who underwent intraoral reconstruction for medium-sized oral cavity defects between 2022 and 2025, using either a facial artery musculomucosal (FAMM) flap or a free flap. Statistical comparisons were made for excision parameters, operative variables, length of hospitalization, and postoperative outcomes. **Results**: Comparison between the FAMM and free flap reconstruction groups revealed no significant differences in patient age, gender, histopathological diagnosis, lesion size, oncological radicality, or functional outcomes related to speech and alimentation. However, FAMM flap reconstruction was associated with significantly reduced operative time (196.7 ± 94.9 min vs. 427.1 ± 129.8 min; *p* < 0.001), representing a 54% reduction in procedure duration. Similarly, the mean hospital stay was 40% shorter in the FAMM group (12.7 ± 6.0 days vs. 21.1 ± 8.0 days; *p* < 0.001). Intensive care unit admission was also markedly less frequent following FAMM flap reconstruction (7.3% vs. 83.3%; *p* < 0.001). **Conclusions**: In cases of small and medium-sized oral cavity defects, reconstruction using FAMM flaps represents a favorable alternative to free flap reconstruction, offering comparable functional outcomes while significantly reducing operative time and length of hospitalization.

## 1. Introduction

Oral squamous cell carcinoma accounts for approximately 90–95% of oral malignancies, followed by minor salivary gland cancers. Surgical treatment typically involves resection of the primary tumor, often accompanied by elective or therapeutic neck dissection and appropriate defect reconstruction usually combined with postoperative radiotherapy [1,2,3]. Reconstruction of the oral cavity remains a major challenge for achieving optimal mechanical, esthetic, and functional rehabilitation after cancer resection. The choice of reconstruction depends on the size and location of the defect (tongue, floor of mouth, buccal mucosa, palate, or gingiva) as well as its components (soft tissue and/or bone). Most defects are managed using local, regional, or free flaps [4,5,6].

Currently, the majority of post-surgical oral cavity defects are reconstructed using free flaps, with radial forearm free flaps (RFFFs) and anterolateral thigh (ALT) flaps representing the most common options [5,7,8]. Microsurgical techniques provide excellent functional and esthetic results, making it possible to reconstruct even the most extensive defects. The introduction of chimeric flaps has further expanded the possibilities, enabling reconstruction after highly extensive resections that were previously avoided due to concerns about the postoperative quality of life [8]. Free flaps are now an essential component of the reconstructive surgeon’s armamentarium. However, these procedures generally require larger surgical teams (often two teams working simultaneously), specialized equipment, expensive sutures and instruments, and a microscope or coupler for venous anastomosis. The costs of microsurgical reconstructions can place a significant financial burden on departments that are funded by national health systems. Using local flaps instead of free flaps for minor reconstructions could significantly reduce the financial impact on surgical departments and hospitals.

Health systems worldwide, in both developed and developing countries, are often burdened by waste, inefficiency, high costs and inadequate care. These issues manifest in various forms, ranging from unnecessary delays to unsafe patient treatment practices [9]. Lean management (LM) is a systematic approach to organizing operations that enhances healthcare quality and safety, improves efficiency, and reduces costs [10]. In adopting lean principles, the key dependent variables include quality of service, cost, and time [10]. Surgical reconstructive procedures should aim to ensure an optimal quality of life, promote proper healing, and minimize the risk of complications, while also reducing operative time to achieve both time and cost savings. The available literature contains only a small number of studies focusing on lean management in the operating theater that do not deal with surgical procedures [11,12].

The aim of this study was to compare the utility, operative time, and costs of oral cavity defect reconstruction using facial artery musculomucosal (FAMM) flaps and free flaps.

## 2. Materials and Methods

A retrospective analysis was conducted of 59 medical records of patients with oral cavity defects requiring primary or secondary flap reconstruction at the Department of Cranio-Maxillofacial Surgery, University Hospital, Cracow, Poland, between January 2022 and August 2025. Patients who underwent bone reconstruction, non-FAMM locoregional flap reconstruction, or had defects larger than 5 cm in diameter were excluded from the study. The present analysis was conducted exclusively on cases involving FAMM flaps and soft tissue free flaps. Due to the fact that defects up to 5 cm in diameter in the oral cavity can be successfully reconstructed with a FAMM flap, a defect diameter of up to 5 cm was used as the cut-off point in the exclusion criteria for soft tissue free flaps. Only RFFFs smaller than 5 cm met the inclusion criteria for the study. ALT and vastus lateralis flaps were excluded due to their size. The choice of reconstructive method was primarily guided by clinical considerations and surgeon preference, taking into account defect size, location, patient comorbidities, and overall operative risk. The process of patient selection is illustrated in the flowchart presented in Figure 1.

The study group comprised 59 patients (32 males and 27 females) with oral cancer who underwent intraoral reconstruction of medium-sized soft tissue defects using either FAMM or free flaps. Patient medical records were reviewed for primary versus secondary reconstruction, histopathological findings, operative time, length of hospital and Intensive Care Unit (ICU) stay, need for tracheostomy or gastrostomy, postoperative complications, recurrence, follow-up, postoperative speech, dietary outcomes, and overall clinical results.

The mean total hospitalization cost (per patient) was calculated as a sum of cost of hospitalization days and cost of operative time. The total cost of ICU stay in both groups was calculated by multiplying the mean number of ICU days by the cost of a single ICU day and by the number of patients who required ICU care. This total ICU cost was then divided by the number of patients in each group to determine the average ICU cost per patient, which was subsequently added to the average total treatment cost for each group.

All costs are expressed in United States dollars (USD), converted from Polish zloty (PLN) at the mean exchange rate for October 2025 (USD 1 = 3.6 PLN). USD was used as the reference (reserve) currency. The unit costs used are based on the institutional actual costs:Operating room (OR) cost per hour: 3500 PLN = 972 USD for one-team surgery (FAMM flap); 4600 PLN = 1278 USD for two-team surgery (free flap).Ward hospitalization cost per day: 1900 PLN = 528 USD per day.ICU hospitalization cost per day: 5600 PLN = 1556 USD per day.

This study was approved by the Institutional Review Board of Jagiellonian University (No. 1072.6120.229.2021). Owing to its retrospective design and the exclusive use of medical records, the review board waived the requirement for patient consent, provided that all personal information remained confidential. All included patients were diagnosed with oral cancer.

Statistical analyses were conducted using IBM SPSS Statistics, version 29.0.2.0 (IBM Corp., Armonk, NY, USA). Pairwise comparisons were assessed using the Mann–Whitney U test when appropriate, and the Chi-square test of independence for categorical data. A *p*-value below 0.05 was considered statistically significant.

## 3. Results

The study cohort comprised 59 patients, with a mean age of 64.0 ± 12.67 years old and a mean body mass index (BMI) of 25.02 ± 4.03 kg/m^2^. A history of tobacco use was reported by 33 patients (55.9%), while 12 (20.3%) reported occasional alcohol consumption, and 4 (6.8%) met the criteria for alcohol dependence. The predominant histopathological diagnosis was squamous cell carcinoma (SCC) (79.7%), followed by mucoepidermoid carcinoma (10.2%), pleomorphic adenoma (8.5%), and adenocarcinoma NOS (1.7%). The most common lesion localizations included the palate (25.4%), tongue (23.7%), and combined involvement of the tongue and floor of the mouth (20.3%). The mean lesion diameter measured 24.91 ± 11.73 mm, with lesions most commonly located on the right side of the oral cavity (54.2%). Among the study cohort, FAMM flap reconstruction was performed in 41 patients (69.5%), including the double-pedicled FAMM (dpFAMM) flap in 73.2% FAMM patients and the island FAMM (iFAMM) flap in 26.8% cases, while 18 patients (30.5%) underwent free flap reconstruction using the RFFF. The majority of procedures were primary reconstructions (83.1%), while secondary reconstructions accounted for 16.95% of cases (Table 1).

When comparing the characteristics of patients who underwent FAMM flap versus free flap reconstruction, no statistically significant differences were observed regarding age, gender, or BMI. In the free flap group, a significantly higher proportion of patients were smokers and alcohol consumers (*p* = 0.04). The number of comorbidities did not differ significantly between the groups. However, a tendency toward a higher comorbidity burden was noted among patients in the RFFF group. This likely reflects the selection of free flaps in patients with more complex medical conditions, although a larger sample size would be required to confirm this observation. No statistically significant differences were observed between the groups with respect to histopathological diagnosis, lesion size, or reconstruction stage (primary vs. secondary). The lesion localization patterns were generally similar. No patients with hard or soft palate defects were included in the free flap group, which influenced the *p*-value in this comparison (Table 2). This is related to the fact that the FAMM flap is the method of choice for palate reconstruction in our department.

The study further evaluated differences between the two reconstructive approaches with respect to surgical procedure characteristics and the corresponding hospital resource utilization (Table 3). The mean operative time was significantly shorter in the FAMM group compared with the free flap cohort (196.7 ± 94.9 vs. 427.1 ± 129.8 min, *p* < 0.001), corresponding to a 54% decrease in procedure duration. Likewise, the FAMM group demonstrated a 40% shorter length of hospital stay (12.7 ± 6.0 vs. 21.1 ± 8.0 days, *p* < 0.001).

The mean total hospitalization cost per patient for FAMM reconstruction was USD 10,404, representing a 56% reduction compared with RFFF reconstruction (USD 23,607, *p* < 0.001). Intensive care unit admission was substantially lower in the FAMM group (7.3% vs. 83.3%, *p* < 0.001). However, when ICU care was required, the duration of ICU stay was marginally longer (4.3 ± 2.6 vs. 2.7 ± 1.6 days, *p* < 0.001). The total ICU stay cost for patients in the FAMM group was 20,211 USD, which was three times lower than the cost in the free flap group (62,318 USD, *p* < 0.001). The mean ICU cost per patient was 493 USD in the FAMM group compared with 3462 USD in the RFFF group (*p* < 0.001) (Figure 2, Figure 3 and Figure 4).

Tracheostomy was performed in a minority of patients in both cohorts, with a lower frequency observed in the FAMM group compared with the free flap group (7.3% vs. 16.7%, *p* = 0.357). Similarly, postoperative enteral nutrition was needed in only a small number of FAMM patients—one case via percutaneous endoscopic gastrostomy (2.4%)—whereas a higher proportion of free flap recipients required enteral support, with two cases via percutaneous endoscopic gastrostomy (11.1%) and one case via surgical gastrostomy (5.6%). However, this difference did not achieve statistical significance.

Regarding surgical outcomes, the majority of patients in both groups achieved R0 resection (FAMM flap: 36/41, 87.8%; free flap: 14/18, 77.8%; *p* = 0.684), with no statistically significant difference in resection margins. Neck dissection was performed unilaterally or bilaterally depending on the clinical indication. In the FAMM flap group, 5 patients (12.2%) underwent unilateral dissection and 22 (53.7%) bilateral, whereas in the free flap group, 1 patient (5.6%) had unilateral and 15 (83.3%) had bilateral dissection (*p* = 0.111 and *p* = 0.386, respectively). Concerning the type of ipsilateral neck procedure, selective neck dissection (SND) predominated in both groups (FAMM: 26/41, 63.4%; free flap: 15/18, 83.3%; *p* = 0.702), with modified radical neck dissection (MRND) performed in a minority of cases. For the contralateral neck, SND was also the most frequent approach (FAMM: 22/41, 53.7%; free flap: 15/18, 83.3%), with no MRND performed on the contralateral side in either group.

Radiotherapy was administered postoperatively in 18 FAMM flap patients (43.9%) and 12 free flap patients (66.7%), with a smaller proportion receiving preoperative radiotherapy (FAMM: 12.2%; free flap: 11.1%; *p* = 0.268). These patients had secondary reconstruction.

Overall, postoperative complications occurred in 8 patients (19.5%) in the FAMM flap group and 7 patients (38.9%) in the free flap group (*p* = 0.116). In the FAMM group, complications included wound dehiscence in 6 patients (14.6%) and wound hematoma in 2 patients (4.9%). In the free flap group, complications were observed as wound dehiscence in 1 patient (5.6%), wound hematoma in 1 patient (5.6%), partial flap necrosis in 4 patients (22.2%), and complete flap necrosis in 1 patient (5.6%). Overall, surgical radicality, type of neck dissection, radiotherapy administration, and complication rates were comparable between the FAMM and free flap cohorts, although there was a trend toward higher rates of partial flap necrosis in the free flap group.

The study also evaluated the functional outcomes of the procedures in both groups with respect to speech and dietary outcomes at >3 months postoperatively (Table 4). To this end, the Functional Effects—Performance Status Scale for Head and Neck Cancer (PSS-HN) [13] was used as follows:**Speech outcomes**100—Speech fully intelligible to all listeners.75—Speech intelligible to most listeners; mild articulation disturbances present.50—Speech intelligible only to persons familiar with the patient, requires listener concentration.25—Speech intelligible only to close family members or caregivers.0—Speech unintelligible or absence of verbal communication ability.
**Dietary outcomes**100—Normal diet without restrictions.75—Able to eat most solid foods; avoids some difficult-to-chew items (e.g., tough meat).50—Soft or semi-liquid (pureed) diet.25—Limited to oral fluid intake only.0—Unable to take food orally—fed via tube (gastrostomy).

Among the 32 patients successfully contacted and assessed in the outpatient clinic (23 in the FAMM group and 9 in the RFFF group), functional outcomes were generally satisfactory for both reconstruction methods (Figure 5). Twenty-five patients (78.1%) rated their post-procedure speech highly, with scores between 75 and 100 points, indicating speech that was intelligible to most listeners with only mild articulation disturbances. In contrast, seven patients (21.9%) scored below 50 points, with speech intelligible only to close family members or caregivers. Regarding postoperative dietary function, 19 patients (59.4%) were able to consume all foods or had only minor restrictions, whereas 40.6% were limited to a soft or pureed diet. The median speech PSS-HN score was 75.0 [IQR 56.25–100.0] in the FAMM group and 75.0 [IQR 75.0–93.75] in the free flap group, with no statistically significant difference (*p* = 0.534). The median nutrition PSS-HN score was 75.0 [IQR 50.0–100.0] for FAMM patients and 50.0 [IQR 50.0–100.0] for free flap patients, also showing no statistically significant difference (*p* = 0.243) (Table 4). These findings indicate that both reconstruction methods resulted in comparable functional outcomes in terms of speech intelligibility and dietary function at more than 3 months postoperatively. However, only 32 of the 54 living patients were assessed for functional outcomes related to speech intelligibility and dietary function. Acknowledging the potential for non-response bias is important, as the patients who were not evaluated may differ systematically from those who participated.

The mean follow-up time was comparable between the FAMM (11.03 ± 12.67 months) and free flap (10.75 ± 9.51 months) groups (*p* = 0.719) (Table 5).

Local recurrence occurred in 2 patients (4.9%) in the FAMM group and 4 patients (22.2%) in the free flap group, with a mean time to recurrence of 19.0 ± 19.8 months and 7.3 ± 2.5 months, respectively (*p* = 0.643). Nodal recurrence was observed in 2 FAMM patients (4.9%) and 4 free flap patients (22.2%), with a mean time to nodal recurrence of 1.50 ± 0.7 months and 4.6 ± 2.9 months, respectively (*p* = 0.064). The need for flap modeling was significantly higher in the FAMM group (20 patients, 48.8%) compared to the free flap group (2 patients, 11.1%; *p* = 0.008). This was connected with the excision and modeling of the flap pedicle in the dpFAMM.

Death due to oral cavity cancer occurred in 3 FAMM patients (7.3%), with a mean time to death of 6.5 ± 4.3 months, and in 2 free flap patients (11.1%), with a mean time to death of 10.0 ± 1.4 months, (*p* = 0.248). Both FAMM and free flap reconstructions appear to provide equivalent oncologic safety and survival. However, FAMM flaps may be associated with a higher need for postoperative flap management.

## 4. Discussion

The application of LM principles in the selection of oral cavity reconstruction methods for post-oncologic surgery patients represents an innovative approach that can enhance clinical decision-making and improve overall quality of care. Implementing LM concepts, such as waste elimination, process standardization, and continuous improvement, may significantly reduce decision-making time and optimize the utilization of hospital resources [14,15]. LM principles can improve efficiency in surgical departments, especially those with limited human resources. One of the primary effects of LM implementation is the organization of the decision-making process through clear criteria for selecting reconstruction methods based on tumor stage, defect size, and the overall condition of the patient. Standardized decision pathways reduce subjectivity in method selection, ensuring reproducible clinical outcomes and minimizing the risk of complications [16,17]. LM also promotes interdisciplinary collaboration. In the context of oral cavity reconstruction, cooperation among surgeons, oncologists, radiotherapeutist, prosthodontists, anesthesiologists and nurses is crucial for achieving optimal functional and esthetic outcomes. Structured communication and clear role allocation reduce errors, increase procedural predictability, and support patient involvement in decision-making, fostering a sense of control and safety [18]. Additionally, the analysis of processes using LM principles improves the efficiency of resource use, including medical materials, surgical equipment, time and human resources. Eliminating redundant or repetitive procedures can lower treatment costs while maintaining high-quality care, which is particularly important in public hospitals with limited resources [19,20].

Reconstruction of oral cavity defects following oncologic resections or trauma remains a significant challenge, both technically and economically. The choice of the reconstructive method should consider not only functional and esthetic outcomes but also treatment costs, hospitalization time, and potential complications. In this study, we compared the costs of treatment in patients undergoing oral cavity reconstruction using the FAMM flap and RFFF. Our findings indicate that the use of the FAMM flap is associated with significantly lower overall treatment costs compared to the RFFF. The reduction in cost is primarily due to shorter operative time, the absence of microsurgical vessel anastomosis, and a shorter postoperative hospital stay. Moreover, the FAMM flap can be harvested and inset under standard operative conditions with one team approach, without the need for highly specialized microsurgical tools, which substantially reduces personnel and equipment expenses [21,22]. FAMM flaps provide a like-for-like mucosal replacement, maintaining the functional and esthetic properties of the oral lining as well as minimal donor site morbidity, as the intraoral harvest results in no external scarring and rapid mucosal healing [6,21]. Moreover, the surgical technique is relatively straightforward. However, one specific aspect of FAMM flap reconstruction that may influence the total treatment cost is the need for a secondary procedure to divide and remodel the flap pedicle. This procedure is typically performed at least four weeks after the initial reconstruction to improve mobility and the contour of the reconstructed area. Although this additional intervention may slightly increase the overall cost of treatment, it can usually be performed under local anesthesia in an outpatient setting, thereby limiting additional hospital-related expenses [6]. This issue can often be avoided with the islanded (iFAMM) variant [23]. Other limitations of FAMM flaps include restricted flap size, possible postoperative scarring, which may lead to trismus, especially after postoperative radiotherapy, and the potential impact of previous facial vessel ligation during neck dissection [6,21,24]. The differences between dpFAMM and iFAMM flaps may influence operative time, flap modeling, and functional outcomes. The dpFAMM flap typically requires a shorter harvesting time due to its straightforward design and limited arc of rotation. However, because of its size and pedicle bulk, dpFAMM often necessitates additional pedicle modeling to optimize intraoral mobility and functional outcomes. In contrast, the iFAMM flap provides a wider arc of rotation and can reach more distant defects. It can be tunnelized under the mandible to reconstruct the tongue or floor of the mouth in patients with full dentition. The iFAMM flap generally requires less postoperative modeling and is usually smaller compared with the dpFAMM flap [6,21,22].

Conversely, the RFFF is regarded as the gold standard for extensive soft tissue reconstructions, especially in cases involving both mucosal and cutaneous components (through-and-through defect). Its advantages include a reliable vascular supply and a favorable healing process, which minimize the risk of flap failure [25,26]. In addition, the provision of sensory reinnervation should be considered, as this has been demonstrated to improve functional outcomes [27]. Furthermore, the RFFF facilitates a two-team approach, incorporating simultaneous harvest and tumor resection, thereby reducing operative time [25,26]. However, these benefits must be balanced against longer operative times, a higher risk of donor site morbidity, and increased costs related to postoperative monitoring and microsurgical care, rehabilitation, hair growth and risk of flap shrinkage, especially after postoperative radiotherapy [28,29].

The economic aspects of oral cavity reconstruction have gained increasing attention, particularly regarding the choice between pedicled (locoregional) and free flaps. Numerous studies indicate that pedicled flaps (FAMM flap, submental island flap-SMIF, supraclavicular artery island flap, and pectoralis major flap) can provide comparable functional and oncologic outcomes to free tissue transfers, including the RFFF, while significantly reducing treatment costs in appropriately selected patients. Paydarfar and Patel were among the first to compare the SMIF with the RFFF in oral cavity reconstruction [30]. The study demonstrated that the submental flap was associated with significantly shorter operative time, shorter hospitalization, and comparable functional outcomes. Their findings suggested that for small and moderate oral defects, local flaps could achieve satisfactory results at lower cost and with less resource utilization. Hu et al., in a systematic review and meta-analysis, confirmed these findings, showing that SMIFs are associated with shorter operation time, decreased hospital stay, and similar complication and flap failure rates compared with free flaps. The cost savings stem mainly from reduced intraoperative duration, simplified postoperative monitoring, and shorter recovery [31]. In addition, Forner et al. performed a direct cost analysis comparing SMIF and RFFF reconstruction in patients with oral cancer and found a substantial reduction in total hospital expenditure in the SMIF group. The savings were attributed to shorter surgical time, less demand for intensive postoperative care, and reduced use of hospital resources, without compromising oncologic safety or functional outcomes [32]. Similarly, Sittitrai et al. reported that pedicled flap reconstruction yielded equivalent functional and esthetic results but with significantly lower complication rates, hospital costs, and length of stay compared to free flaps [33]. Katna et al. reached analogous conclusions, noting that pedicled flaps remain cost-effective alternatives in settings with limited microsurgical capacity or financial resources [34]. A recent narrative review by Louizakis et al. has reinforced the view that while free tissue transfer remains the gold standard for large or complex composite defects, locoregional flaps are increasingly preferred for small and medium-sized reconstructions due to their reduced operative burden, shorter anesthesia time, and lower overall cost [35]. Kozin et al. and Zhang et al. also demonstrated that the supraclavicular artery island flap provides outcomes comparable to fasciocutaneous free flaps, including the RFFF, while significantly reducing operative duration and hospitalization costs [36,37]. Furthermore, studies comparing the pectoralis major pedicled flap with free flaps have shown similar findings [38,39]. While free flaps offer superior contour and pliability, pedicled options remain viable for resource-constrained environments, balancing acceptable function and lower costs. On the other hand, the multicenter Taiwanese study by Liao et al. reported higher risk of inadequate margins resection as well as worse overall survival and disease-specific survival among patients treated with local flaps [40]. On the other hand, in the presented cohort difference between margin resection, overall survival (OS) and disease-specific survival (DSS) in FAMM flap and free flap reconstruction was not observed, emphasizing that clinical outcomes need not be compromised when choosing cost-effective reconstructive strategies.

Our observations are consistent with previous studies emphasizing that the FAMM flap represents a valuable alternative to free flaps in small and medium-sized oral cavity defects [41,42]. Similar to the findings of Ibrahim et al. and Joseph et al., our data demonstrate that the FAMM flap is associated with significantly shorter operative time, reduced duration of hospitalization, and lower rate of ICU admission, when compared with free flap techniques (Table 6). Notably, the mean operative time in our cohort (FAMM: 196.7 min vs. RFFF: 427.1 min) was considerably shorter than that reported by Ibrahim et al. (FAMM: 432 min vs. RFFF: 534 min) [41]. Conversely, in the study by Joseph et al., the reported mean operative time for iFAMM flaps (56 min) differs substantially from that observed in our study (196 min for FAMM flaps) [42]. This discrepancy likely results from differences in methodological definitions of operative time, as Joseph et al. measured only the duration of flap harvest and reconstruction, whereas in our study, the total operative time also included the period required for neck dissection.

Furthermore, the difference in hospital stay duration was more pronounced in our study (FAMM: 12.7 days vs. RFFF: 21.1 days, *p* < 0.001), while Ibrahim et al. reported no significant difference in this parameter, suggesting that our patients experienced a faster postoperative recovery and earlier discharge when reconstructed with the FAMM flap [41]. Consistent with the findings of Joseph et al., our analysis confirmed a markedly lower need for ICU admission among patients undergoing FAMM flap reconstruction [42]. However, the duration of ICU stay in our study, when required, was slightly longer in the FAMM group.

Postoperative complication rates in our series (19.5% for FAMM vs. 38.9% for RFFF) followed the general trend observed in previous studies, underscoring the lower morbidity associated with local flaps [41,42]. Functional outcomes, including speech intelligibility and swallowing capacity, remained comparable between the two reconstructive methods, corroborating the previously described functional adequacy of the FAMM flap [41,42]. Our findings reinforce and extend the conclusions of Joseph et al. and Ibrahim et al., confirming that the FAMM flap is a reliable, technically less demanding, and cost-effective reconstructive option for medium-sized oral cavity defects. It provides satisfactory functional and esthetic outcomes while significantly reducing operative time, hospital stay, and the overall treatment burden compared with microvascular free flap reconstruction.

The multicenter analysis by Vaira et al., encompassing 615 patients (390 FAMM and buccinator myomucosal flaps vs. 225 free flap reconstructions), provides the most comprehensive evidence to date supporting the oncologic safety and clinical efficacy of the FAMM flap [43]. Their findings demonstrated non-inferior oncologic outcomes compared with free flaps, with comparable five-year progression-free survival (69.8% vs. 66.2%), OS (77.9% vs. 73.5%), and DSS (95.3% vs. 94.2%). Local and regional recurrence rates were likewise similar between groups, confirming that preservation of the facial vessels during selective neck dissection does not compromise oncologic radicality [43]. Our results are consistent with these observations, further substantiating the reliability of the FAMM flap. In our cohort, both local and nodal recurrence rates were markedly lower in the FAMM group (4.9%) compared with the free flap group (22.2%), while DSS remained favorable. The relatively large sample size of the FAMM group in our study further reinforces the safety and effectiveness of the FAMM technique, not only in terms of shorter operative time and hospitalization, but also with regard to long-term prognosis and disease control. Collectively, these findings, in concordance with the multicenter data from Vaira et al., confirm that the FAMM flap represents a safe, efficient, and reliable reconstructive option that ensures excellent functional and esthetic outcomes while maintaining survival and recurrence rates comparable to those achieved with microvascular free flaps [41,42,43].

Economic analyses have demonstrated that, with appropriate patient selection and surgical planning, the FAMM flap can achieve comparable functional and esthetic outcomes while significantly reducing treatment costs [41,42]. However, the choice of reconstructive technique must always be individualized. Patients with extensive defects, limited local tissue availability, or limited blood supply require free flap reconstruction, despite higher costs, which may remain the only viable option [5]. Additionally, a comprehensive cost-effectiveness analysis should consider not only the direct costs of hospitalization, but also long-term functional outcomes, quality of life and the potential need for revision surgeries [6,41,42,43,44].

## 5. Study Limitations

This study is limited by its retrospective, single-center design and small sample size, which may affect generalizability and statistical power. Additionally, the analyzed cohorts differed substantially in sample size, with the free flap group representing a markedly smaller cohort compared with the FAMM group. The smaller sample size in the free-flap cohort increases the risk of type II error, meaning that true differences may not reach statistical significance. Another limitation of this study is the potential for selection bias in the choice between FAMM flaps and free flap reconstruction. Because the reconstructive method was determined by surgeon preference and individualized clinical judgment rather than a standardized preoperative protocol. As a result, the two groups may not be fully comparable, which could affect the interpretation of outcome differences between reconstructive techniques. Another limitation of the study is the relatively short follow-up period, which may not capture long-term variability in function. It should be emphasized, however, that FAMM flaps are generally stable in size and do not undergo significant shrinkage over time or after adjuvant radiotherapy, unlike soft tissue free flaps, which can exhibit greater variability in volume and function over several years. The number of complications observed was low, which may have limited the statistical power to detect significant differences. A further limitation of the study is that no multivariable or propensity-adjusted analyses were performed in order to account for baseline differences between the groups, which may have introduced confounding in the comparison of outcomes. For example, patients in the free flap group had a statistically higher rate of comorbidities, alcohol dependence, and cigarette smoking, factors that could potentially increase hospital stay and the need for ICU admission. The analysis included only the mean cost of hospital stay, which was calculated based on the average duration of surgery, hospitalization days, and ICU stay. The economic analysis was restricted to direct treatment costs, as LM tools are not implemented in our hospital, preventing assessment of indirect costs and overall resource utilization. Despite its benefits, LM implementation in oncologic care faces several challenges. It requires cultural change, full team engagement, and investment in training and monitoring tools. Furthermore, the individualization of cancer patient treatment demands flexibility in standardization, which can limit the full applicability of LM principles in every case. Future research should focus on measuring the impact of LM implementation on clinical outcomes, patient satisfaction, and treatment costs. Specifically, prospective, randomized studies comparing traditional decision-making models with LM-based approaches would provide objective evidence of the effectiveness of this method in post-oncologic oral cavity reconstruction.

## 6. Conclusions

The FAMM flap constitutes a reliable and cost-effective option for the reconstruction of small to medium-sized oral cavity defects, providing satisfactory functional and esthetic outcomes while reducing operative time, hospitalization, ICU stay and overall treatment costs compared to free flaps such as RFFF. From an LM perspective, the optimization of workflow, the reduction in unnecessary operative time, and the minimization of hospital resource utilization could further enhance the cost-effectiveness of FAMM flap reconstruction. From an economic standpoint, a personalized reconstructive strategy is recommended: pedicled flaps, such as the FAMM flap, are indicated for minor to moderate defects, while free flaps are advised for large or complex reconstructions.

## Figures and Tables

**Figure 1 jcm-14-08760-f001:**
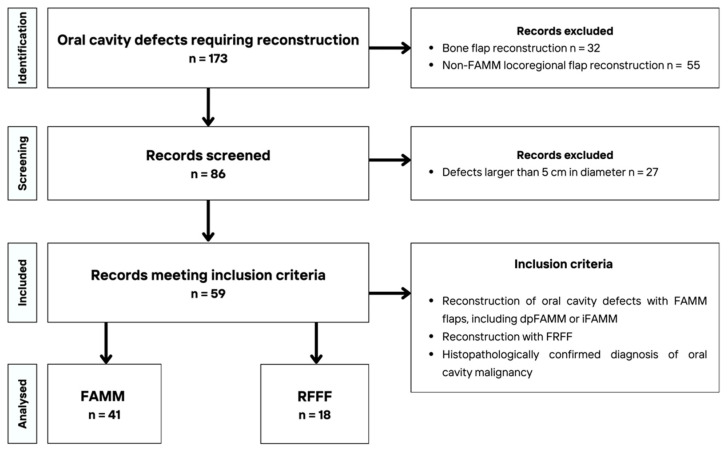
Flowchart of patient selection in study.

**Figure 2 jcm-14-08760-f002:**
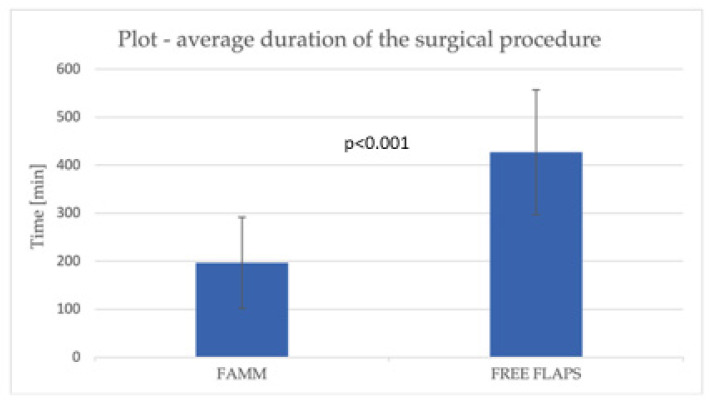
A chart illustrating the average surgery duration.

**Figure 3 jcm-14-08760-f003:**
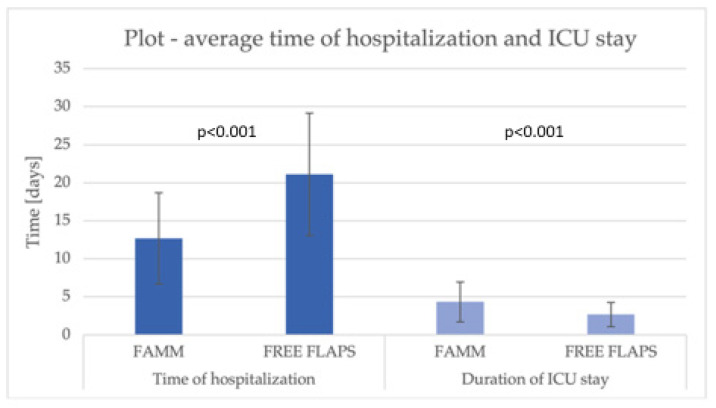
A chart illustrating the average length of hospital and ICU stay.

**Figure 4 jcm-14-08760-f004:**
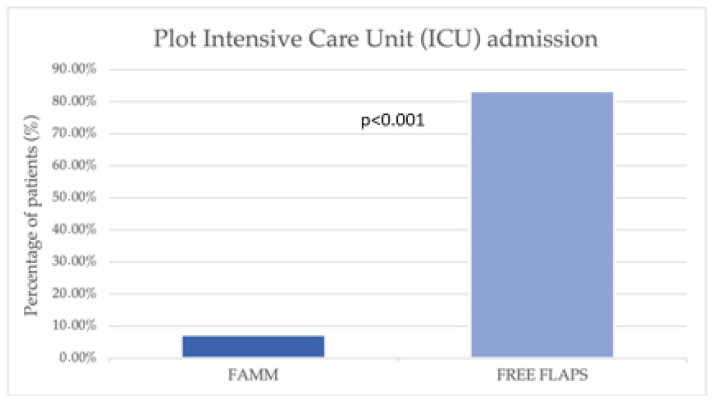
A chart illustrating the frequency of ICU admission after surgery.

**Figure 5 jcm-14-08760-f005:**
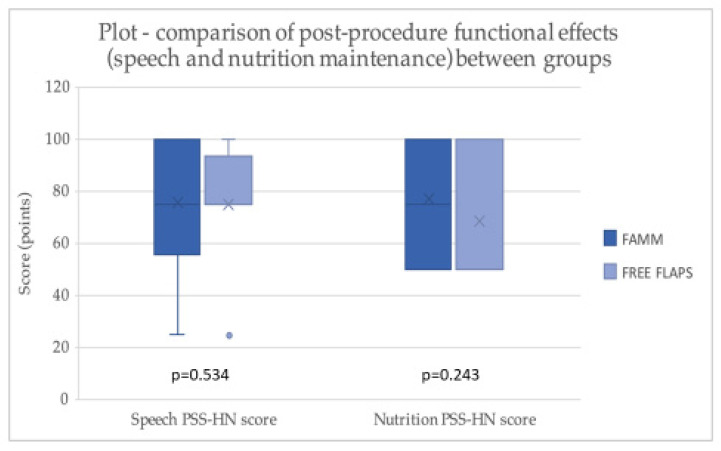
A chart illustrating post-procedure functional effects within groups.

**Table 1 jcm-14-08760-t001:** The general characteristics of the study population.

**Total number of patients (n, %)**	59 (100%)
**Age, [years], mean (SD)**	64.00 (±12.67)
**Gender (n, %)**	
Female (n, %)	27 (45.76%)
Male (n, %)	32 (54.24%)
**BMI [kg/m^2^], mean (SD)**	25.02 (±4.03)
**Smoking history (n, %)**	
Non-smoker	26 (44.07%)
Smoker	33 (55.93%)
**Alcohol consumption (n, %)**	
Non-drinker	43 (77.97%)
Occasional drinker (less than 3 times per week)	12 (20.34%)
Alcohol dependence syndrome	4 (6.78%)
**Histopathological diagnosis (n, %)**	
Squamous cell carcinoma	47 (79.66%)
Mucoepidermoid carcinoma	6 (10.17%)
Pleomorphic adenoma	5 (8.48%)
Adenocarcinoma NOS	1 (1.69%)
**Lesion location (n, %)**	
Tongue	14 (23.73%)
Floor of the mouth	8 (13.56%)
Both tongue and floor of the mouth	12 (20.34%)
Hard or soft palate	15 (25.42%)
Retromolar triangle and gingiva	10 (16.95%)
**Lesion site (n, %)**	
Right	32 (54.24%)
Left	20 (33.90%)
Midline	7 (11.86%)
**Lesion measurement [mm], mean (SD)**	24.91 (±11.73)
**Surgical technique (n, %)**	
FAMM flap	41 (69.49%)
dpFAMM	30 (73.17%)
iFAMM	11 (26.83%)
Free flap	18 (30.51%)
RFFF	18 (100.00%)
**Type of procedure (n, %)**	
Primary reconstruction	49 (83.05%)
Secondary reconstruction	10 (16.95%)

**Table 2 jcm-14-08760-t002:** Comparison of the characteristics of the FAMM flap and free flap groups.

Study Group	FAMM	FREE FLAPS	*p* Value
**Total number of patients (n, %)**	41 (100%)	18 (100%)	-
**Age, [years], mean (SD)**	65.32 (±13.21)	61.82 (±10.79)	0.123
**Gender (n, %)**			0.07
Female (n, %)	22 (53.66%)	5 (27.78%)	
Male (n, %)	19 (46.34%)	13 (72.22%)	
**BMI [kg/m^2^], mean (SD)**	24.75 (±4.27)	26.54 (±3.43)	0.736
**Comorbidities count (n, %)**			0.07
One or less	22 (53.66%)	5 (27.78%)	
More than one	19 (46.34%)	13 (72.22%)	
**Smoking history (n, %)**			0.038
Non-smoker	21 (51.22%)	5 (27.78%)	
Smoker	20 (48.78%)	13 (72.22%)	
**Alcohol consumption (n, %)**			0.032
Non-drinker	29 (70.73%)	14 (77.78%)	
Occasional drinker (<3 times/week)	11 (26.83%)	1 (5.56%)	
Alcohol dependence syndrome	1 (2.44%)	3 (16.67%)	
**Histopathological diagnosis (n, %)**			-
Squamous cell carcinoma	19 (46.34%)	18 (100%)	
Mucoepidermoid carcinoma	2 (4.88%)		
Pleomorphic adenoma	5 (12.20%)		
Adenocarcinoma NOS	1 (2.44%)		
**Lesion location (n, %)**			<0.001
Tongue	9 (21.95%)	5 (27.78%)	
Floor of the mouth	6 (14.63%)	2 (11.11%)	
Both tongue and floor of the mouth	9 (21.95%)	3 (16.67%)	
Hard or soft palate	15 (36.59%)	0 (0.00%)	
Retromolar triangle and gingiva area	2 (4.88%)	8 (44.44%)	
**Lesion site (n, %)**			0.425
Right	24 (58.64%)	8 (44.44%)	
Left	13 (31.71%)	7 (38.89%)	
Midline	4 (9.76%)	3 (16.67%)	
**Lesion measurement [mm], mean (SD)**	23.53 (±12.39)	28.52 (±9.53)	0.148
**Type of procedure (n, %)**			0.141
Primary reconstruction	36 (87.80%)	13 (72.22%)	
Secondary reconstruction	5 (12.20%)	5 (27.78%)	

**Table 3 jcm-14-08760-t003:** Comparison of the surgical procedures’ characteristics.

Procedure Type	FAMM FLAPS	FREE FLAPS	*p* Value
**Duration of the surgical procedure [min], mean (SD)**	196.71 (±94.92)	427.11 (±129.76)	<0.001
**Time of hospitalization [days], mean (SD)**	12.68 (±6.00)	21.11 (±8.04)	<0.001
**Intensive Care Unit (ICU) admission (n, %)**	3 (7.32%)	15 (83.33%)	<0.001
**Duration of ICU stay [days], mean (SD)**	4.33 (±2.62)	2.67 (±1.59)	<0.001
**Tracheostomy performed (n, %)**	3 (7.32%)	3 (16.67%)	0.357
**Postoperative enteral nutrition (n, %)**	1 (2.44%)	3 (16.67%)	0.203
Percutaneous endoscopic gastrostomy	1 (2.44%)	2 (11.11%)	
Surgical gastrostomy	0 (0%)	1 (5.56%)	
**Resection type (n, %)**			0.684
R0	36 (87.80%)	14 (77.78%)	
R1	5 (12.20%)	4 (22.22%)	
**Neck lymph node dissection (n, %)**	27 (65.85%)	16 (88.89%)	0.111
unilateral	5 (12.20%)	1 (5.56%)	0.386
bilateral	22 (53.66%)	15 (83.33%)	
**Type of ipsilateral procedure (n, %)**			0.702
SND	26 (63.41%)	15 (83.33%)	
MRND	1 (2.44%)	1 (5.56%)	
**Type of contralateral procedure (n, %)**			-
SND	22 (53.66%)	15 (83.33%)	
MRND	0 (0%)	0 (0%)	
**Radiotherapy (n, %)**			0.268
Postoperative	18 (43.90%)	12 (66.67%)	
Preoperative	5 (12.20%)	2 (11.11%)	
**Complications occurrence (n, %)**	8 (19.51%)	7 (38.89%)	0.116
Wound dehiscence	6 (14.63%)	1 (5.56%)	
Wound hematoma	2 (4.88%)	1 (5.56%)	
Partial flap necrosis	0 (0.00%)	4 (22.22%)	
Complete flap necrosis	0 (0.00%)	1 (5.56%)	

**Table 4 jcm-14-08760-t004:** Comparison of post-procedure functional effects (speech and nutrition maintenance) between groups.

Study Group	FAMM	FREE FLAPS	*p* Value
**Speech PSS-HN score**			
Median [IQR]	75.00 [56.25–100.00]	75.00 [75.00–93.75]	0.534
**Nutrition PSS-HN score**			
Median [IQR]	75.00 [50.00–100.00]	50.00 [50.00–100.00]	0.243

**Table 5 jcm-14-08760-t005:** Long-term outcomes of FAMM and free flaps procedures.

Study Group	FAMM	FREE FLAPS	*p* Value
**Follow up time [months], mean (SD)**	11.03 (±12.67)	10.75 (±9.51)	0.719
**Local recurrences (n, %)**	2 (4.88%)	4 (22.22%)	0.643
Time to local recurrence, [months], mean (SD)	19.00 (±19.80)	7.25 (±2.50)	
**Nodal recurrences (n, %)**	2 (4.88%)	4 (22.22%)	0.064
Time to nodal recurrence, [months], mean (SD)	1.50 (±0.71)	4.60 (±2.88)	
**Need for flap modeling (n, %)**	20 (48.78%)	2 (11.11%)	0.008
**Disease specific survival (n, %)**	3 (7.32%)	2 (11.11%)	0.248
Time to death [months], mean (SD)	6.50 (±4.27)	10.00 (±1.41)	

**Table 6 jcm-14-08760-t006:** Comparison of studies results evaluating FAMM flap versus free flap techniques.

Results	Joseph et al., 2020 [42]	Ibrahim et al., 2021 [41]	Gontarz et al., 2025
**Number of patients**	40 (20 iFAMM, 20 FCFF)	31 (13 FAMM, 18 RFFF)	59 (41 FAMM, 18 RFFF)
**Mean age [years]**	iFAMM 51.5 vs. FCFF 44.7 (*p* = 0.38)	FAMM 66 vs. RFFF 62 (*p* = 0.306)	FAMM 66 vs. RFFF 61.8 (*p* = 0.123)
**Female/male ratio**	not given (majority male)	5:8 in FAMM; 8:10 in RFFF	12:10 in FAMM, 4:10 in RFFF
**Surgical method**	islanded facial artery myomucosal (iFAMM) flap; fasciocutaneous free flap (FCFF)	FAMM RFFF	FAMM RFFF
**Operating time (mean)**	iFAMM 56.5 min vs. FCFF 150.5 min (*p* < 0.001)	FAMM 432 min vs. RFFF 534 min (*p*= 0.002)	FAMM 196.7 min vs RFFF 427.1 min, (*p* < 0.001)
**Hospital stay (mean)**	iFAMM 7.5 days vs. FCFF 9.4 days (*p* = 0.45)	No significant difference (*p* = 0.717)	FAMM 12.7 days vs. RFFF 21.1 days (*p* < 0.001)
**ICU stay (mean)**	iFAMM 1-day vs. FCFF 3.2 days (*p* < 0.001)	Not specified (rare ICU use for FAMM)	ICU admission significantly lower in FAMM group (7.3% vs. 83.3%, *p* < 0.001); However, the duration of ICU stay marginally longer in FAMM (4.3 vs. 2.7 days, *p* < 0.001)
**Need for tracheostomy**	iFAMM 0 vs. FCFF 14 cases (*p* < 0.001)	Both groups 100% tracheostomy	Tracheostomy performed in the minority of patients in both groups
**Complications**	iFAMM 5 minor complications vs. FCFF 5 minor complications, no flap loss in both groups	FAMM 1 minor (7.7%) vs. RFFF 15 complications in 10 patients (55%) (*p* = 0.008)	FAMM 8 minor complications (19.5%) vs. RFFF 7 minor complications (38.9%) in (*p* = 0.116). Complete flap necrosis in 1 RFFF patient, no flaps loss in FAMM group
**Functional outcomes**	Speech & swallowing similar (*p* > 0.1); Esthetics better for iFAMM (VAS 8.4 vs. 6.0)	Speech & swallowing similar (*p* > 0.05)	Speech & swallowing similar (*p* > 0.05)
**Economic cost**	FCFF ≈ 30% more expensive than iFAMM	FAMM CAD 24,188 vs. RFFF CAD 35,247 (–31% difference, *p* = 0.109)	FAMM USD 10,404 vs. RFFF USD 23,607 (–56% difference, *p* < 0.001)
**Overall conclusion**	The iFAMM flap was associated with shorter operative and ICU stay times, reduced hospitalization time, and greater cost efficiency, while providing superior esthetic outcomes and lower donor site morbidity.	FAMM flaps are associated with lower costs, shorter OR time, similar functional outcomes and a tendency to lower complication rates.	FAMM flap constitutes a reliable and cost-effective option, with satisfactory functional and esthetic outcomes while reducing operative time, hospitalization, and overall treatment costs.

## Data Availability

The data presented in this study are available on request from the corresponding author. Data are not publicly available due to privacy.

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
