# Peer review of "Lean Management in Medium-Sized Oral Cavity Defect Reconstruction: Facial Artery Musculomucosal Flaps Versus Free Flaps"

_jcm, 2025, doi:10.3390/jcm14248760_

Round 1
Reviewer 1 Report
Comments and Suggestions for Authors
Dear Authors,
Thank you for the opportunity to review your manuscript entitled “Lean Management in Medium-Sized Oral Cavity Defect Reconstruction: FAMM Flaps Versus Free Flaps.” The topic is timely and relevant, and the study offers meaningful insights into cost-effective reconstructive strategies, particularly through the lens of Lean Management. The manuscript is generally well structured and presents a comprehensive comparison between FAMM flaps and radial forearm free flaps. I commend the authors for the clear presentation of quantitative results, detailed methodology, and thoughtful discussion of LM principles.
While the study is promising, I believe the manuscript could be strengthened with the following recommendations:
1. Introduction
-
The introduction provides a solid overview of oral cavity reconstruction, but the transition to Lean Management appears somewhat abrupt. Consider more clearly articulating why Lean Management specifically motivates your comparison of FAMM versus free flaps. A stronger conceptual link would improve narrative continuity.
-
It may be helpful to explicitly define the gap in the literature at the end of the introduction and outline how your study addresses it.
2. Methods
-
The retrospective design is well explained; however, the rationale for selecting the 5-cm defect cut-off could be elaborated, as this choice directly impacts comparability between groups.
-
Please clarify whether preoperative selection criteria for FAMM vs free flap were standardized or based on surgeon preference, as this introduces potential selection bias.
-
Consider providing more detail regarding the calculation of operative costs, specifically how one-team versus two-team surgeries were allocated in your institution’s workflow.
3. Results
-
Tables are clear and informative. However, the distribution of flap types creates a substantial imbalance between groups (41 vs 18). Please comment on how this influences statistical power.
-
The significantly higher proportion of smokers and alcohol users in the free flap group (Table 2)
jcm-4007607-peer-review-v1
merits additional discussion, as these factors may influence complication rates and functional outcomes.
-
In the functional outcomes section, only 32 of 59 patients could be contacted. Discussing potential non-response bias would strengthen the interpretation of these results.
4. Discussion
-
The discussion provides an excellent literature review; however, some external studies are described with great detail that could be condensed to maintain focus on your findings.
-
The application of Lean Management principles is described conceptually. It would strengthen the paper to explicitly link LM principles to specific findings, for example:
-
reduced operative time (Figures 2–4)
jcm-4007607-peer-review-v1
-
lower ICU use
-
simplified workflow due to the one-team approach
-
-
Consider expanding on whether LM principles were actively applied in your department or whether the analysis is retrospective and interpretive.
5. Limitations
-
The limitations section is well written, but it may help to more clearly acknowledge:
-
the heterogeneity of flap types within the FAMM group (iFAMM vs dpFAMM),
-
the lack of adjustment for baseline differences such as smoking/alcohol exposure,
-
and the exclusion of ICU costs, which may distort total cost comparisons.
-
6. Conclusions
-
The conclusion is consistent with the results; however, consider tempering statements regarding cost-effectiveness, given the excluded ICU costs, non-randomized design, and sample size imbalance.
7. Minor suggestions
-
Proofread for minor typographical inconsistencies (e.g., spacing, punctuation, and table alignment).
-
Ensure acronyms (FAMM, RFFF, LM) are used consistently after first definition.
-
Some figure captions could benefit from more descriptive detail (e.g., sample size, meaning of error bars).
Author Response
- Introduction
- The introduction provides a solid overview of oral cavity reconstruction, but the transition to Lean Management appears somewhat abrupt. Consider more clearly articulating why Lean Management specifically motivates your comparisonof FAMM versus free flaps. A stronger conceptual link would improve narrative continuity.
- It may be helpful to explicitly define the gap in the literatureat the end of the introduction and outline how your study addresses it.
Thank you for your constructive feedback. Following a review, changes were made to the introduction in line with the suggestions made.
- Methods
- The retrospective design is well explained; however, the rationale for selecting the 5-cm defect cut-offcould be elaborated, as this choice directly impacts comparability between groups.
A comprehensive explanation has been provided.
Please clarify whether preoperative selection criteria for FAMM vs free flap were standardized or based on surgeon preference, as this introduces potential selection bias.
Thank you for this important comment. Preoperative selection between FAMM flaps and free flaps was not based on a formal, standardized protocol. Instead, the choice of reconstructive method was primarily guided by clinical considerations and surgeon preference, taking into account defect size, defect location, patient comorbidities, and overall operative risk. We acknowledge that this approach may introduce selection bias, and we have added a statement to the manuscript to clarify this limitation.
- Consider providing more detail regarding the calculation of operative costs, specifically how one-team versus two-team surgeries were allocated in your institution’s workflow.
Appropriate changes have been made to the materials and methods used to assess treatment costs. In addition, data on the costs of treatment in the ICU have been added.
- Results
- Tables are clear and informative. However, the distribution of flap types creates a substantial imbalance between groups (41 vs 18). Please comment on how this influences statistical power.
A comprehensive explanation has been included in the study limitations section.
- The significantly higher proportion of smokers and alcohol users in the free flap group (Table 2) merits additional discussion, as these factors may influence complication rates and functional outcomes.
- In the functional outcomes section, only 32 of 59 patients could be contacted. Discussing potential non-response biaswould strengthen the interpretation of these results.
Thank you for this comment. The appropriate clarification has been incorporated into the Results section.
- Discussion
- The discussion provides an excellent literature review; however, some external studies are described with great detail that could be condensed to maintain focus on your findings.
Thank you for this observation. We aimed to highlight that not only the FAMM flap but also other reconstructive options may serve as cost-effective alternatives to free flaps for medium-sized oral cavity defects. These external studies were included to provide context and support for this broader perspective.
- The application of Lean Management principles is described conceptually. It would strengthen the paper to explicitly link LM principles to specific findings, for example:
- reduced operative time (Figures 2–4)
- lower ICU use
- simplified workflow due to the one-team approach
- Consider expanding on whether LM principles were actively applied in your departmentor whether the analysis is retrospective and interpretive.
Thank you for this valuable suggestion. LM principles were discussed conceptually in the manuscript. We have clarified in the revised text that several observed outcomes, such as reduced operative time, lower ICU utilization, and a simplified workflow enabled by the one-team approach, are fully consistent with LM concepts. We also specify that LM principles were not formally implemented as part of an institutional improvement initiative. Rather, the analysis is retrospective and interpretive, demonstrating how the observed differences between reconstruction methods align with LM principles and illustrating the potential for their future structured application in reconstructive decision-making.
- 5. Limitations
- The limitations section is well written, but it may help to more clearly acknowledge:
- the heterogeneity of flap typeswithin the FAMM group (iFAMM vs dpFAMM),
- the lack of adjustment for baseline differencessuch as smoking/alcohol exposure,
- and the exclusion of ICU costs, which may distort total cost comparisons.
Thank you for all of the suggestions. Proper changes in the study limitation based on other reviewers suggestions have been made.
- Conclusions
- The conclusion is consistent with the results; however, consider tempering statements regarding cost-effectiveness, given the excluded ICU costs, non-randomized design, and sample size imbalance.
Thank you for this insightful comment. The concerns regarding cost-effectiveness, the retrospective study design, and the imbalance in sample size, have been addressed and acknowledged in the study limitations section.
- Minor suggestions
- Proofread for minor typographical inconsistencies (e.g., spacing, punctuation, and table alignment).
Typographical changes were made.
- Ensure acronyms (FAMM, RFFF, LM) are used consistently after first definition.
Changes were made.
- Some figure captions could benefit from more descriptive detail (e.g., sample size, meaning of error bars).
Thank you for the constructive feedback. All changes to the manuscript are marked in red.
Reviewer 2 Report
Comments and Suggestions for Authors
comments
- add more about the pros and cons of the different contrusction methods in simple summary
- report statistical significance in the charts
- give a better terminology of post-procedure functional effects (speech and nutrition
maintenance) - nice and well-presented discussion. Generally a well described with many references and nice comparisons of different techniques and correct associations with purpose of treatment and other factors such as cost and post-op timings
Author Response
Add more about the pros and cons of the different reconstruction methods in simple summary
Thank you for the constructive feedback. In accordance with the reviewers’ suggestions, a brief summary of the reconstructive methods has been added.
Report statistical significance in the charts
P-values have been added to the charts.
Give a better terminology of post-procedure functional effects (speech and nutrition
maintenance)
The terminology has been improved. All changes to the manuscript are marked in red.
Reviewer 3 Report
Comments and Suggestions for Authors
Your study included 59 patients who underwent reconstruction of medium-sized oral cavity defects using either the FAMM flap (41 patients) or the radial forearm free flap (18 patients). You compared operative time, hospitalization, ICU use, complications, functional results, recurrence, and overall cost.
Your findings showed that the FAMM flap required much shorter operative time, shorter hospitalization, very low ICU use, and far lower overall cost, while still maintaining similar functional and oncologic outcomes compared with free flaps. Both groups had similar speech and dietary outcomes on the PSS-HN scale.
Comments about the methodology
The retrospective design is appropriate for the question, and your inclusion and exclusion criteria are clearly shown. The patient-selection flowchart is easy to follow. Your statistical tests are appropriate, although the group size difference (41 vs 18) does make some comparisons harder to interpret. The functional assessment using PSS-HN is a strong point, since it is a validated tool.
Some baseline differences between groups, such as smoking, alcohol use, and comorbidities, could affect healing, complications, and recurrence. These deserve a clearer explanation or adjustment.
Strengths of the study
1. Your cost and operative-time comparisons are very strong and clearly presented.
2. The oncologic outcomes are reassuring and support the safety of using the FAMM flap.
3. The use of PSS-HN for assessing speech and dietary function strengthens the functional comparison.
4. The discussion is well informed and connects your findings with several recent and relevant studies.
Revisions I would recommend
1. Please explain more clearly how differences like smoking, alcohol use, and comorbidity burden may have influenced the results. These factors independently affect postoperative healing, complications, and recurrence. Adding an adjusted analysis or at least a more detailed discussion would make your conclusions stronger.
2. It would help to briefly describe the dpFAMM and iFAMM variations and how they influence operative time, flap modeling, or functional outcomes. Readers may not be familiar with these technical differences.
3. Regarding the cost analysis, ICU costs were excluded because of the imbalance in ICU admissions. However, for real-world interpretation, ICU costs carry major weight. It would strengthen the paper if you could discuss how excluding these costs might bias the total cost comparison, or provide a basic estimate of how ICU involvement might shift the financial picture.
This study brings meaningful insights and supports a practical, cost-conscious approach to reconstruction. It is well written and has real clinical value. If you address the points above, your manuscript will be even clearer and more rigorous.
Author Response
- Please explain more clearly how differences like smoking, alcohol use, and comorbidity burden may have influenced the results. These factors independently affect postoperative healing, complications, and recurrence. Adding an adjusted analysis or at least a more detailed discussion would make your conclusions stronger.
Thank you for your constructive comments. This issue has been addressed in the study limitations section.
- It would help to briefly describe the dpFAMM and iFAMM variations and how they influence operative time, flap modeling, or functional outcomes. Readers may not be familiar with these technical differences.
Thank you for this suggestion. We have added a brief description of difference between the dpFAMM and iFAMM flap variations in discussion section.
- Regarding the cost analysis, ICU costs were excluded because of the imbalance in ICU admissions. However, for real-world interpretation, ICU costs carry major weight. It would strengthen the paper if you could discuss how excluding these costs might bias the total cost comparison, or provide a basic estimate of how ICU involvement might shift the financial picture.
Thank you for your comment. We have included the calculated cost of ICU stay and the difference between the FAMM and free flap groups. The overall cast has been updated.
Reviewer 4 Report
Comments and Suggestions for Authors
This study compares FAMM flaps (n=41) and radial forearm free flaps (RFFF; n=18) for reconstruction of medium-sized soft-tissue oral cavity defects (<5 cm). The authors analyze operative time, ICU use, length of stay, costs, complications, oncologic outcomes, and PSS-HN–based functional results.
GENERAL COMMENTS: Despite the clinical relevance of comparing FAMM and free flaps for medium-sized oral cavity defects, the strength of the conclusions is limited by several methodological constraints. The retrospective single-center design, small and imbalanced cohorts (41 FAMM vs 18 RFFF), and clear confounding by indication (including higher smoking and alcohol rates and a trend toward greater comorbidity in the free flap group) reduce internal validity. The cost analysis is restricted to direct ward and OR costs and excludes ICU costs and all downstream or indirect expenses, weakening the economic argument. Functional outcomes were available only for a subset of 32 patients, with no multivariable adjustment for baseline differences, and oncologic follow-up is relatively short. Therefore, the message that FAMM flaps offer equivalent oncologic and functional results at lower cost should be interpreted as hypothesis-generating rather than definitive.
Materials & methods:
How were patients allocated to FAMM vs RFFF? Please detail the decision algorithm or criteria.
Did the authors consider any multivariable or propensity-adjusted analysis to address baseline differences?
How were costs calculated (institutional actual costs, standard tariffs, or estimates)?
Why were ICU costs excluded, given that ICU use differed markedly between the groups?
How were PSS-HN questionnaires administered, and how many patients per group completed them?
Can the authors provide further stratification of tumor stage (T and N categories) by reconstruction type?
Discussion:
Temper the causal language and emphasize that results are associative due to selection bias.
Clarify that lean management principles are used conceptually and were not implemented as part of an institutional improvement pathway.
Explicitly acknowledge confounding by indication as a major limitation in comparing operative time, ICU use, and LOS.
Author Response
GENERAL COMMENTS:
Despite the clinical relevance of comparing FAMM and free flaps for medium-sized oral cavity defects, the strength of the conclusions is limited by several methodological constraints. The retrospective single-center design, small and imbalanced cohorts (41 FAMM vs 18 RFFF), and clear confounding by indication (including higher smoking and alcohol rates and a trend toward greater comorbidity in the free flap group) reduce internal validity. The cost analysis is restricted to direct ward and OR costs and excludes ICU costs and all downstream or indirect expenses, weakening the economic argument. Functional outcomes were available only for a subset of 32 patients, with no multivariable adjustment for baseline differences, and oncologic follow-up is relatively short. Therefore, the message that FAMM flaps offer equivalent oncologic and functional results at lower cost should be interpreted as hypothesis-generating rather than definitive.
Thank you for your constructive comments. All of the issues raised by the reviewer have been addressed and incorporated into the revised manuscript.
Materials & methods:
How were patients allocated to FAMM vs RFFF? Please detail the decision algorithm or criteria.
Proper information has now been added to the methodology section.
Did the authors consider any multivariable or propensity-adjusted analysis to address baseline differences?
No, we did not perform multivariable or propensity-adjusted analyses. The study was primarily descriptive and aimed at comparing costs between the FAMM and free flap groups. We acknowledge that baseline differences between groups could influence the results, and we have added a statement in the limitations section to address this potential confounding factor.
How were costs calculated (institutional actual costs, standard tariffs, or estimates)?
The financial costs were calculated on the basis of the actual costs of the institution. The methodology has been updated to include the necessary information.
Why were ICU costs excluded, given that ICU use differed markedly between the groups?
Thank you for your comment. We have included the calculated cost of ICU stay and the difference between the FAMM and free flap groups. The overall cast has been updated.
How were PSS-HN questionnaires administered, and how many patients per group completed them?
Thank you for this suggestion. The Results section has been updated.
Can the authors provide further stratification of tumor stage (T and N categories) by reconstruction type?
Thank you for the suggestion. However, further stratification of tumor stage (T and N categories) by reconstruction type was not within the scope or aim of this study, which focused primarily on comparing outcomes, costs, and complications between FAMM and free flap reconstructions. We believe that including this information would add unnecessary detail and could detract from the clarity of the manuscript.
Discussion:
Temper the causal language and emphasize that results are associative due to selection bias.
Thank you for the suggestion. We have revised the manuscript to emphasize that the observed differences are associative, as the choice of reconstruction was influenced by surgeon preference and clinical factors, which may introduce selection bias.
Clarify that lean management principles are used conceptually and were not implemented as part of an institutional improvement pathway.
Thank you for the comment. We would like to clarify that this publication emphasizes the potential for implementing lean management principles in the selection of reconstructive procedures. The concepts are discussed conceptually and were not applied as part of a formal institutional improvement pathway.
Explicitly acknowledge confounding by indication as a major limitation in comparing operative time, ICU use, and LOS.
The study limitation section has been expanded to include additional information. All changes to the manuscript are marked in red.
Round 2
Reviewer 4 Report
Comments and Suggestions for Authors
I appreciate your efforts in addressing the comments and improving the manuscript accordingly.